# The Extraction Process, Separation, and Identification of Curcuminoids from Turmeric *Curcuma longa*

**DOI:** 10.3390/foods12214000

**Published:** 2023-11-01

**Authors:** Gal Slaček, Petra Kotnik, Azra Osmić, Vesna Postružnik, Željko Knez, Matjaž Finšgar, Maša Knez Marevci

**Affiliations:** 1Faculty of Chemistry and Chemical Engineering, University of Maribor, SI-2000 Maribor, Slovenia; gal.slacek@um.si (G.S.); petra.kotnik@um.si (P.K.); azra.osmic@um.si (A.O.); vesna.postruznik@um.si (V.P.); zeljko.knez@um.si (Ž.K.); matjaz.finsgar@um.si (M.F.); 2Faculty of Medicine, University of Maribor, Taborska 8, SI-2000 Maribor, Slovenia

**Keywords:** turmeric, curcumin, total phenols, proanthocyanidins, antioxidants

## Abstract

Turmeric *Curcuma longa* is a well-known spice with various health benefits, attributed primarily to curcumin. Soxhlet extraction, cold maceration, ultrasound-assisted extraction (UAE), and supercritical fluid extraction were performed, and the content of total phenols, proanthocyanidins, and antioxidants was analysed by UV/VIS spectrophotometry. High-performance liquid chromatography–tandem mass spectrometry (HPLC-MS/MS) was employed to identify and quantify the curcumin content. Supercritical extracts had the highest total phenolic content (538.95 mg GA/100 g material), while the Soxhlet extracts had the highest content of proanthocyanidins (4.77 mg PAC/100 g of material). Extracts obtained by UAE and supercritical extraction have the highest antioxidant potential. Antioxidant activity measured by 2,2-diphenyl-1-picrylhydrazyl (DPPH^•^) was 64.27% and 1750.32 mg Trolox per g dry weight by 2,2-azinobis(3-ethylbenzothiazoline 6 sulphonic acid) (ABTS^+•^) for the extract obtained by supercritical extraction. The UAE resulted in the highest amount of curcumin (1.91 mg curcumin/g material). A kinetic study showed that extraction yield in supercritical extracts decreased with increasing temperature and that the content of isolated curcumin was inversely proportional to solvent-to-feed ratio (S/F). The present study has confirmed that turmeric is an excellent source of antioxidants, such as curcumin, that play an important role in reducing cellular stress by neutralising free radicals.

## 1. Introduction

The steep rise in inflammatory and cancerous diseases has led to a massive increase in research on natural antioxidants or natural materials rich in antioxidants. One of the promising natural materials rich in antioxidants is turmeric. Turmeric is a source of antioxidants, vitamins (B3, B6, C, and E), volatile components (turmerone, atlantone, and zingiberene), and mineral elements (Ca, Mg, P, Fe, Zn, and Na) which have a positive effect on health and various disease treatments [1,2,3].

Turmeric is derived from the rhizomes of the *Curcuma longa* plant, which belongs to the ginger family (Zingiberaceae). Curcuminoids, which are fat-soluble polyphenols, give the plant its orange colour and have many favourable properties for human health. The main curcuminoid in the plant is curcumin [4]. Curcumin was first isolated in 1815 by Vogel and Pelletier, but it took 98 years, when in 1913 Lampe and Miłobędzka validated its chemical structure (Figure 1) (Figure 1) [5,6]. Curcumin is a hydrophobic compound and is soluble in organic solvents (methanol, ethanol, acetone, benzene, etc.). Curcumin is found in nature in two forms: in the keto monoole and diketone form. Jayaprakasha et al. [7] confirmed that curcumin is the most potent antioxidant of all curcuminoids. Turmeric also contains dimethoxycurcumin and bis-dimethoxycurcumin as main compounds [8]. The proportion of all three curcuminoids depends on geographical conditions and harvesting. Curcumin (80%) presents the largest proportion of curcuminoids, followed by dimethoxycurcumin (17%) and bis-dimethoxycurcumin (3%) [9,10].

The extraction procedures for extracting curcuminoids from Curcuma longa were already reported [11,12,13]. Several extraction techniques (Soxhlet, microwave-assisted extraction, cold maceration, supercritical fluid extraction, and ultrasound-assisted extraction (UAE)) for optimising curcuminoid content in water or fat-soluble extracts different solvents (acetone, water, ethanol, triacetin, and diacetin) were used.

The water- and fat-soluble extracts of turmeric and its active components exhibit antioxidant activity comparable to that of vitamins C and E. Recent studies have demonstrated turmeric’s anti-inflammatory and anti-cancer activity [14,15]. However, the use of curcumin as an active medicinal ingredient is limited by its poor bioavailability, as it is insoluble in water and rapidly degrades in the intestine [16,17]. Proper treatment and processing of the plant are crucial to isolate the biologically active or medicinal components of turmeric for human consumption. Antioxidants, such as curcumin, play an important role in reducing cellular stress by neutralising free radicals [18].

In this work, different extraction techniques (Soxhlet, UAE, cold maceration) and supercritical fluid extraction were performed to study the influence of extraction technique and process parameters (time, pressure, and temperature) on the content of bioactive components in turmeric extracts. Supercritical fluid extraction has been applied as a sustainable method for the extraction of the target compounds. Herein, the simultaneous investigation of the influence of extraction technique and process parameters (time, pressure, and temperature) on the content of bioactive components in turmeric extracts was performed followed by subsequent statistical analysis. The supercritical fluid extraction allowed facile separation of the solvent from the extracts, and thus, despite the lower mass yield, the recovery of phenolic compounds was the highest. Additionally, the data on the curcumin extraction kinetics have been provided. The mass yield and curcumin content in the extract obtained by supercritical extraction at 25 MPa, 40, 50, and 60 °C as a function of solvent-to-feed ratio have been studied. The optimal process parameters and extraction time for the highest recovery of curcumin have been determined, which will be valuable for future studies.

Using different extraction methods (Soxhlet extraction, UAE, supercritical extraction, and cold maceration) the optimal extraction conditions of turmeric for the isolation of high contents of bioactive components (antioxidants, proanthocyanidins, and total phenolics) have been determined. Increasing the temperature and pressure parameters had a favourable effect on the increase in the extraction yield and the content of bioactive components of the individual samples.

The aim of the manuscript is to compare different extraction techniques and the analytical performance of extracts by high-performance liquid chromatography coupled with tandem mass spectrometry (HPLC-MS/MS). In addition, the evaluation of experimental extraction results by statistical analyses was performed.

## 2. Experimental

### 2.1. Materials

*Curcuma longa* was supplied by Alfred Galke GmbH (Samtgemeinde, Bad Grund, Germany). The material was ground and dried. The moisture content of the turmeric powder was determined using a halogen moisture analyser HX204 (Mettler Toledo, Greifensee, Switzerland). Ethanol (CAS 64-17-5, Sigma-Aldrich Chemie GmbH, Steinheim, Germany) with purity ≥ 99.9% was used as a solvent for conventional extractions and 2,2-azinobis(3-ethylbenzothiazoline 6 sulphonic acid) (ABTS^+•^) analysis. Methanol (CAS reg.nNo. 67-56-1, Sigma-Aldrich Chemie GmbH) with purity ≥ 99.9% and 2,2-diphenyl-1-picrylhydrazyl (DPPH^•^) (Sigma-Aldrich Chemie GmbH) with purity ≥ 97.0% were used for DPPH^•^ analysis. CO_2_ (CAS reg. no. 124-38-9) was purchased from MESSER (MG-Ruše, Slovenia), with a purity of 99.99%. All analytical standards (curcumin, product no. 08511, with purity ≥ 98.0%; bisdemethoxycurcumin, product no. 90,594 with purity ≥ 95.0%; demethoxycurcumin, product no. 90,593 with purity ≥ 95.0%), ABTS^+•^ (product no. 194430), diammonium salt, potassium persulfate (K_2_S_2_O_8_, product no. 216224), (±)-6-hydroxy-2,5,7,8-tetramethylchromane-2-carboxylic acid (Trolox, 97.0%), Folin–Ciocalteu’s reagent (FC, product no. 1.09001), anhydrous sodium carbonate (Na_2_CO_3_, product no. 223530), and ferrous sulfate heptahydrate (FeSO_4_·7H_2_O, product no. 215422) were purchased from Sigma-Aldrich. Hydrochloric acid (HCl, 37.0%) and n-butanol with purity ≥ 99.5% were supplied by Merck (Darmstadt, Germany) and used for proanthocyanidins content.

### 2.2. Moisture Content

A halogen moisture analyser HX204 was used to determine the moisture content of turmeric powder, following the Elbl et al. [19] method with some modifications. To conduct the analysis, 3 g of turmeric powder was weighed in a crucible. The thermogravimetric principle was used to measure the sample. The samples were heated to 100 °C until a constant weight was obtained, and the weight difference was recorded. The measurement was performed in triplicate, and the amount of moisture is expressed as mass % with standard deviation.

### 2.3. Extraction Methods

Soxhlet extraction, UAE, cold maceration, and supercritical extraction were performed. The solvents used were ethanol for conventional extractions and supercritical CO_2_ in combination with ethanol as co-solvent for supercritical extractions. The extraction time was determined based on previous reports [9,20,21,22,23,24,25,26] and previous extraction experiences. Each extraction procedure was performed in triplicate.

#### 2.3.1. Soxhlet Extraction

Soxhlet extraction was carried out in a water bath at the solvent’s boiling point under ambient air pressure (laboratory conditions). The procedure has been described in detail previously [27]. Briefly, 25.00 g material was extracted with 150 mL of solvent (ethanol) using Soxhlet extraction apparatus. After 3 h of extraction, the solvent was evaporated using vacuum rotavapor (BÜCHI Rotavapor R-114 and BÜCHI Vacuum Controller B-721, BÜCHI Labortechnik AG, Flawil, Switzerland) at 40 °C, and the mass of the extract was determined gravimetrically. The extraction yield (*η*) was calculated by using Equation (1). The extract was stored in a freezer until the spectrophotometric and HPLC-MS/MS analyses were carried out.
(1)$$\eta (wt.\%)=\frac{m_{extract}}{m_{raw\;material}}\times 100,$$

#### 2.3.2. UAE

UAE was carried out at temperatures up to 40 °C. The extraction process has been described in detail previously [27]. Briefly, 25.00 g of material was extracted with 150 mL of solvent (ethanol) using an ultrasound bath (VEVOR, Rancho Cucamonga, CA, USA). The extraction was carried out at a frequency of 40 kHz. After 1.5 h of extraction, the extraction solution was filtered through filter paper (MN-751, Macherey-Nagel, Düren, Germany) and evaporated using rotavapor at 40 °C. The yield of extraction was calculated by using Equation (1). The extract was stored in a freezer until the spectrophotometric and HPLC-MS/MS analyses were carried out.

#### 2.3.3. Cold Maceration

Cold maceration was carried out on a magnetic stirrer (TEHTNICA, Železniki, Slovenia) at 450–500 rpm and room temperature. Briefly, 25.00 g of material was extracted with 150 mL of solvent (ethanol). After 3 h of stirring, the solution was filtered (MN-751) to separate the solid material from the extract solution. The extract solution was evaporated using rotavapor at 40 °C. The yield of extraction was calculated by using Equation (1). The extract was stored in a freezer until the spectrophotometric and HPLC-MS/MS analyses were carried out.

#### 2.3.4. Supercritical Extraction

The procedure employed in this work was described previously [28]. A high-pressure autoclave (Figure 2) filled with 25.00 g of turmeric powder was immersed in a water bath at a constant temperature of 60 °C and pressure of 25 MPa for a duration of 120 min. During the extraction process, ethanol was pumped via a high-performance liquid chromatography (HPLC) pump at a flow rate of 2 mL/min to isolate polar and non-polar components. The pressurised CO_2_ in the autoclave was pumped from the gas cylinder using a high-pressure pump (model 260D, ISCO syringe pump, Lincoln, NE, USA) and was constant throughout the experiment. Separation and expansion of supercritical CO_2_ from ethanol and extract were carried out in a glass trap at atmospheric pressure and room temperature. After the extraction was completed, the co-solvent was further evaporated from the extraction solution using a rotavapor. The yield of extraction was calculated by using Equation (1). The extract was stored in a freezer until the spectrophotometric and HPLC-MS/MS analyses were carried out.

### 2.4. Spectrophotometric Analysis

Antioxidant capacity, total phenolic content, and proanthocyanidins content were determined using a UV/VIS spectrophotometer (Cary 50 Scan, Varian, Surrey, England). The total phenolic content in the extracts was determined using the Folin–Ciocalteu method [29]. Proanthocyanidins content was determined using the UV spectrophotometric method [30] based on acid hydrolysis and colour formation. The antioxidant activity was evaluated quantitatively by scavenging 2,2-diphenyl-1-picrylhydrazyl (DPPH^•^) and 2,2-azinobis(3-ethylbenzothiazoline-6-sulphonic acid) (ABTS^+•^) radicals [31,32]. The extract solution was prepared using 10 mg of extract, which was diluted in 10 mL of methanol.

#### 2.4.1. Determination of Total Phenols

The total phenols were determined using a Folin–Ciocalteu (FC) reagent, as described previously [33]. Briefly, the extract solution (0.5 mL) was mixed with 2.5 mL of FC reagent, which had been diluted 1:10 with distilled water. The resulting mixture was neutralised with 2 mL of Na_2_CO_3_ solution (7.5%, *w*/*v*), and the preparation time was not more than 2 min. The samples were thermostated in a water bath for 5 min at 50 °C and then cooled down. For the control sample, 0.5 mL of distilled water was used instead of the extract. The absorbance was measured at a wavelength of 760 nm. The total phenolic content was expressed as mg of gallic acid (GA) per 100 g of material.

#### 2.4.2. Determination of Proanthocyanidins

The proanthocyanidins content was determined by a hydrolysis reaction in a reaction mixture containing n-butanol and concentrated hydrochloric acid [33]. To prepare the sample, 1 mL of the extract solution with a concentration of 5 mg/mL was used. Then, 10 mL of FeSO_4_·7H_2_O (77 mg/500 mL) in a mixture of HCl and n-butanol (2:3) was added to the prepared solution. The mixture was incubated for 15 min in a water bath at 95 °C. After incubation, the samples were cooled to ambient temperature, and the absorbance was measured at a wavelength of 540 nm. The content of proanthocyanidins was expressed in mg PAC per 100 g of material.

#### 2.4.3. ABTS^+•^ Method

The antioxidant activity of obtained extracts was determined using ABTS^+•^ method [34]. A 7 mmol/L solution of ABTS^+•^ (solid ABTS^+•^) and 2.5 mmol/L solution of K_2_O_8_S_2_ (solid K_2_O_8_S_2_) were prepared in ultrapure water. Ultrapure water was obtained from Elga PURELAB Classic system (ELGA LabWater, High Wycombe, United Kingdom). The working solution was prepared by mixing ABTS^+•^ and K_2_O_8_S_2_ solutions in a 1:1 volume ratio and left in a dark room. The working solution was diluted with ethanol until the absorbance 0.70 at wavelength 734 nm was obtained. The method protocol was to mix 3.950 mL of the ABTS^+•^ solution with 50 µL of the sample (either a standard Trolox solution or a crude extract solution) and incubate it for 30 min in the dark at room temperature. The absorbance of the solution was then measured at 734 nm. A blank solution contained only the ABTS^+•^ solution and ethanol. The radical-scavenging activity was calculated by Equation (2), where *A*_blank_ is the absorbance of the blank solution, and *A*_sample_ is the absorbance of the sample solution. Antioxidant activity was expressed as mg Trolox equivalent per g of dry weight (mg Trolox/g DW).
(2)$$\%Inhibition=\left(\frac{A_{blank}-A_{sample}}{A_{blank}}\right)\times 100,$$

#### 2.4.4. DPPH^•^ Method

The antioxidant activity of obtained extracts was determined using the DPPH^•^ method [31]. The extracts (10 mg) were dissolved in methanol solvent (10 mL). The extract was dissolved by stirring the solution and by exposure to the ultrasound bath for a few minutes. A 0.06 mmol/L solution of DPPH^•^ was prepared of solid DPPH^•^ and methanol. For the samples, 77 µL of the extract solution was mixed with 3 mL of the prepared DPPH^•^ solution in dark bottles, and the mixture was incubated for 15 min at room temperature in a dark room. The absorbance of the solutions was measured at the wavelength of 517 nm. The radical-scavenging activity was calculated by Equation (3), where *A*_0_ is the absorbance of the control solution (*t* = 0 min), and *A*_15_ is the absorbance of the sample solution (*t* = 15 min). The results are presented as % of inhibition at concentration 1 mg extract per mL of methanol.
(3)$$\%Inhibition=\left(\frac{A_{0}-A_{15}}{A_{0}}\right)\times 100,$$

### 2.5. High-Performance Liquid Chromatography–Tandem Mass Spectrometry

The concentration of active components in the turmeric extracts was determined by HPLC-MS/MS using Agilent 1200 HPLC and Agilent 6460 JetStream triple quadrupole mass spectrometer (Agilent Technologies, Santa Clara, CA, USA). The Agilent Poroshell EC-C18 column (2.7 µm particle size, 100 × 2.1 mm ID) was employed for chromatographic analysis. The analysis was carried out at 35 °C using gradient elution. The quantification was performed based on the calibration curve methodology.

Mobile phase A consisted of ultrapure water containing 0.1% formic acid, and mobile phase B was acetonitrile containing 0.1% formic acid (elution gradients: 0 min 50% B, 5 min 85% B, 7 min 85.0% B, 8 min 50.0% B to 10 min). The mass spectrometer was operated in negative ionisation mode with optimised parameters for nitrogen as carrier gas at the temperature of 300 °C and a flow rate of 5 L/min, sheath gas temperature of 250 °C, sheath gas flow of 11 L/min, and capillary and nozzle voltages at 3500 V and 500 V, respectively. The multiple-reaction monitoring (MRM) mode was applied to determine curcumin based on ion transitions from *m/z* 367.1 to 217.0 and 173.0 with collision energy 4 and 12 V, respectively. Corresponding chromatogram and mass spectrum are shown in Figure 3.

### 2.6. Statistical Analysis

The differences in the bioactive activity of the extracts were examined with statistical tests using the R programming language version 4.1.1. and RStudio version 2022.02.03. Data distributions were analysed with Shapiro–Wilk statistical tests. Differences in bioactive activity among normally distributed variables were examined using analysis of variance (ANOVA), whereas the Kruskal–Wallis nonparametric statistical test was used to analyse differences among nonnormally distributed variables. Post hoc Tukey’s test was performed for normally distributed variables, while post hoc Dunn’s test was performed for nonnormally distributed variables. Significant differences between groups were marked on bar plots with different letters. For cases with no significant differences, the same letters were used.

## 3. Results and Discussion

### 3.1. Moisture Content

The moisture content in the turmeric powder was 9.15 ± 0.53% (*w*/*w*), which is comparable to the results obtained by Paulluci et al. [35] and is within the acceptable range specified by the European Medicines Agency [36], which states that values below 13.1% are acceptable. The residual moisture content in the turmeric powder is an indicator of the processing and preservation methods used and can impact the chemical and microbiological stability of the final product. Each measurement of moisture has been performed in triplicate.

### 3.2. Extraction

Different extraction methods were performed to obtain extracts from Curcuma powder. Figure 4 shows the extraction yields (*η*) for different extraction methods. Among all these extraction methods, Soxhlet extraction showed the highest *η* (10.27 wt.%), followed by supercritical extraction (8.67 wt.%), cold maceration (7.62 wt.%), and UAE (6.83 wt.%). Temperature, as well as solvent density, influences the extraction yield. With increasing temperature at constant density, the extraction yield increases, and with increasing solvent density, the extraction yield increases at constant temperature. Differences in extraction yields between methods were compared using ANOVA, which showed significant differences between extract groups (F(3) = 21.88, *p* < 0.001). Sahne et al. [9] reported lower *η* using UAE (3.92 wt.%) and Soxhlet (6.90 wt.%) extraction compared to *η* obtained in this work (6.83 wt.% for UAE and 10.27 wt.% for the Soxhlet extraction). Compared to our experimental work, the authors used different parameters, such as time and solvent, for the extraction methods.

For Soxhlet extraction, Marin et al. [37], compared to the work herein, used five times less material (5 g), added the same amount of ethanol, and extracted for 6 h. Obtained *η* were 1.8% lower compared to what was obtained in this work (10.27 wt.%). Perko et al. [27] had very similar supercritical extraction parameters (20–30 MPa, 60 °C) to our experimental conditions. The authors did not use a co-solvent, which resulted in a lower yield (5.2%) compared to our experimental results, where the use of a co-solvent allowed for the extraction of both non-polar and polar components, resulting in a higher yield of 8.7 wt.%.

Therefore, we can conclude that the yield % of extraction is influenced by different factors such as the amount of material used, the type of solvent [38], contact time [39], and the extraction method [40]. Additionally, temperature [41] is a crucial variable that can impact the efficiency of extraction, as both methods, Soxhlet and supercritical, have higher efficiencies where the temperature is higher than room temperature.

### 3.3. Spectrophotometric Analysis

#### 3.3.1. The Content of Total Phenolic Compounds in Extracts

Total phenols in turmeric extracts obtained by different extraction methods are shown in Figure 5a. The results are shown as a mass of gallic acid in mg per 100 g of material. The supercritical method yielded the highest content of total phenolic components, i.e., 538.95 mg GA/100 g of material. The lowest total phenolic content was determined using the cold maceration extraction (178.42 mg GA/100 g of material). When comparing the phenolic content of the extracts between extraction methods, the Kruskal–Wallis statistical test confirmed significant differences between the extracts (χ(3) = 10.385, *p* = 0.016).

#### 3.3.2. The Content of Proanthocyanidins in Extracts

Figure 5b presents the fraction of proanthocyanidins in turmeric extracts. The results are shown as a mass of proanthocyanidins in mg per 100 g of material (mg PAC/100 g material). Turmeric extracts were obtained by different extraction methods. All the extraction methods used, apart from the UAE, resulted in a similar number of proanthocyanidins. The highest proportion of proanthocyanidins was obtained by Soxhlet extraction (4.77 mg PAC/100 g material), while the lowest proportion of proanthocyanidins was obtained by UAE (2.71 mg PAC/100 g material). ANOVA was used for comparing proanthocyanidin content between extraction methods and confirmed significant differences between extract groups (F(3) = 75.92, *p* < 0.001).

#### 3.3.3. Antioxidant Activity

Figure 5c shows that the sample prepared by the supercritical extraction method using ethanol as co-solvent resulted in the highest antioxidant activity (64.27% inhibition determined by the DPPH^•^ method), whereas the sample prepared by UAE resulted in the lowest antioxidant activity (33.41% inhibition determined by DPPH^•^ method). Fernández-Marín et al. [37] reported that the Soxhlet method resulted in lower inhibition (49.0%) compared to our study (51.52% inhibition). The extraction methods, with a degree of freedom of three, were analysed for their statistical differences using the Kruskal–Wallis statistical test. The test confirmed significant differences between the antioxidant activity of the extracts depending on the extraction method for the DPPH^•^ method of the antioxidant assay (χ(3) = 10.385, *p* = 0.016)

Figure 5d shows that there were significant differences between the samples, confirmed with the Kruskal–Wallis statistical test (χ(3) = 10.385, *p* = 0.016). The highest proportion of antioxidant activity by the ABTS^+•^ method was determined by the UAE method (1750.32 mg Trolox/g DW), while the lowest proportion of antioxidant activity was obtained by the supercritical method (1216.87 mg Trolox/g DW).

There was a visible discrepancy in the antioxidant activity of the samples determined by the two different methods. There could be several reasons. One of them is the different solubility in the two solvents. In the case of DPPH^•^, the samples were dissolved in methanol, whereas in the case of ABTS^+•^, they were dissolved in ethanol. Certain bioactive compounds may not be soluble in the reaction media and may not exert radical-scavenging activities. More likely is the reaction mechanism. ABTS^+•^ and DPPH^•^ both act as stable free radicals but have different reaction mechanisms. ABTS^+•^ is oxidised by antioxidants, resulting in a colour change from green to blue. DPPH^•^, on the other hand, is reduced by antioxidants, resulting in a decolourisation of the purple solution [42,43].

The results are in contrast when comparing the two methods used, DPPH^•^ and ABTS^+•^, to determine antioxidant activity. In the DPPH^•^ method, the sample obtained by the supercritical method had the highest antioxidant activity, and the sample obtained by the UAE method had the lowest antioxidant activity, whereas, in the ABTS^+•^ method, the sample obtained by UAE had the highest antioxidant activity, and the sample obtained by supercritical extraction had the lowest antioxidant activity. The high content of antioxidant activity in the ABTS^+•^ method shows how potently antioxidant turmeric is, as, despite the high dilution of the sample, a considerable proportion was still present in the solution. Based on Figure 5c,d, it can be concluded that the DPPH^•^ method is the most suitable for compounds extracted by supercritical CO_2_ and ethanol, while the ABTS^+•^ method is more suitable for the analysis of extracts prepared with ethanol by UAE (polar components).

### 3.4. HPLC-MS/MS Analysis of Curcumin

Figure 6 presents the curcumin content in the extracts obtained by different extraction methods. The results are given as milligrams of the component per g of material. Figure 6 shows that the extract obtained by UAE contains the highest content of curcumin (1.91 mg curcumin/g material), while there are no significant differences between the other extraction methods (supercritical, cold maceration, Soxhlet), whereas the contents of curcuminoid are somewhat lower. The reason for the high content is most likely the ultrasound waves breaking down the cell walls and causing an intense secretion of curcumin and solubility of compounds in ethanol. For curcumin content using different extraction methods, the Kruskal–Wallis statistical test was performed, but significant differences between extraction methods could not be confirmed (χ(3) = 7.051, *p* = 0.0702).

### 3.5. Kinetics for the Extracts Obtained with Supercritical Fluids

To determine the effect of temperature on extraction efficiency, a study was performed at the temperature range between 40 and 60 °C at constant solvent and co-solvent flow at 10 °C intervals. The solvent-to-feed ratio (S/F) ratio was calculated for samples taken after 15, 30, 60, and 120 min. The increase in temperature had a negative effect on extraction yield, as shown in Figure 7. At the pressure of 25 MPa, the extraction yield decreased with increasing temperature, which can be attributed to a decrease in solvent density. The highest extraction yield was obtained after 120 min at 40 °C, and it was 11.16 wt.%.

For the same samples from the study shown in Figure 7, the release of curcumin as a function of S/F ratio was analysed. Figure 8 shows that the content of isolated curcumin was inversely proportional with S/F ratio. Figure 8 shows that the extract obtained at 60 °C had higher curcumin contents than extracts obtained at lower temperatures. At an S/F ratio of around 3 and a temperature of 60 °C, the sample with the highest curcumin content (1.14 mg curcumin/g material) was obtained.

## 4. Discussion

To summarise, the highest extraction yield was obtained by Soxhlet extraction (10.27 wt.%), followed by supercritical extraction and cold maceration, and the lowest yield was obtained by UAE (6.83 wt.%). DPPH^•^ and ABTS^+•^ assays resulted in different values for the antioxidant ability of the extracts. It is assumed that differences between DPPH^•^ and ABTS^+•^ radical-scavenging activities can be ascribed to reaction mechanism. When DPPH^•^ was used, the sample prepared by the supercritical extraction method using ethanol as a co-solvent had the highest antioxidant activity, with 64.27% inhibition. Since DPPH^•^ is a method applicable to extracts with more phenolic compounds, especially water-soluble ones, the inhibition is consequently higher in extracts with a higher content of phenolic compounds, which is in the extract after supercritical extraction. The lowest total phenolic content was measured by cold maceration (178.42 mg GA/100 g material) and showed 55.58% inhibition using DPPH^•^ method. Not only phenolic compounds, but also proanthocyanidins, are components that affect the level of antioxidant activity in the extract. Thus, extracts containing a higher content of proanthocyanidins show a higher % of inhibition when using DPPH^•^ method. This can also be observed in the extract obtained by Soxhlet extraction, where the content of phenolic compounds is low (278.29 mg GA/100 g material and high in proanthocyanidins (4.77 mg PAC/100 g material). The extract obtained by UAE contained 191.94 mg GA/100 g material and 2.71 mg PAC/100 g material, which were the lowest values among the extracts. The second antioxidant activity method used was ABTS^+•^. Based on the ability of antioxidant compounds, the DPPH^•^ and ABTS^+•^ have different reaction mechanisms. The DPPH^•^ ability of free radical scavenging of a compound is based on the compound’s ability to donate hydrogen atoms. Meanwhile, the ABTS^+•^ ability of antioxidant compounds is based on the ability of antioxidant compounds to stabilise free radical compounds by donating proton radicals [44]. As a result, the DPPH^•^ and ABTS^+•^ provide different information about free radical and antioxidant activity [45]. In the case of the UAE extract, the differences between the antioxidant activity by DPPH^•^ (Figure 5c) and by ABTS^+•^ (Figure 5d) confirmed that assumption. As the extract obtained by UAE contained the highest curcuminoid content (1.91 mg curcumin/g material), and the other extraction methods (supercritical, cold maceration, Soxhlet) did not show significant differences in curcuminoid content when comparing both, the differences in antioxidative activity of those extracts represent minor components such as phenolic acids and proanthocyanidins. Studies have shown a correlation between antioxidant activity by DPPH^•^ and total phenolic content in many plants [46], and this has been shown to be the case for turmeric. At the same time, the content of curcumin (Figure 6) had influence on higher antioxidative activity by ABTS^+•^ in extracts with low content of phenolic compounds and proanthocyanidins.

## 5. Conclusions

The aim of this study was to obtain *Curcuma longa* extracts using different extraction methods (Soxhlet extraction, UAE, supercritical extraction, and cold maceration) and to determine the optimal extraction method for the isolation of bioactive components from turmeric (antioxidants, proanthocyanidins, and total phenolics) with high antioxidative value. The main component extracted was curcumin (analysed by HPLC-MS/MS). The content of other components in turmeric extracts was determined by spectrophotometric analysis. Ethanol proved to be an excellent solvent in all extraction methods used to isolate valuable components from turmeric powder.

The study of kinetics was performed with pressurised CO_2_ extraction. For samples taken after 15, 30, 60, and 120 min, extraction yield was calculated, and curcumin content was measured as a function of solvent-to-feed ratio (S/F). Increasing the temperature and pressure had a favourable effect on the increase in the extraction yield and the content of bioactive components in the individual samples.

The present study has confirmed that turmeric is an excellent and potent antioxidant containing a high content of total phenolics and curcuminoids. Because of all these properties, there is high potential to develop applications to facilitate the uptake of the extract into the human body and to monitor its release.

## Figures and Tables

**Figure 1 foods-12-04000-f001:**
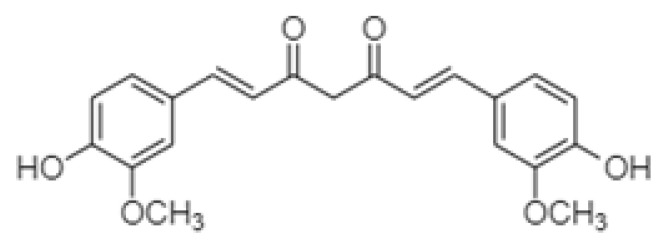
Chemical structure of curcumin.

**Figure 2 foods-12-04000-f002:**
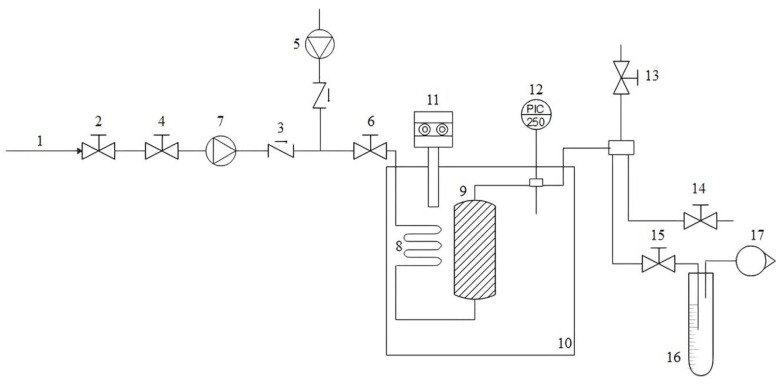
High-pressure extraction apparatus; 1—CO_2_ gas supply, 2,4,6—inlet valves, 3—one-way valve, 5—HPLC pump, 7—high-pressure pump, 8—heating coil, 9—autoclave, 10—thermostated bath, 11—temperature indicator and regulator, 12—manometer, 13,14—outlet valves, 15—extract release valve, 16—sampling trap, 17—rotameter.

**Figure 3 foods-12-04000-f003:**
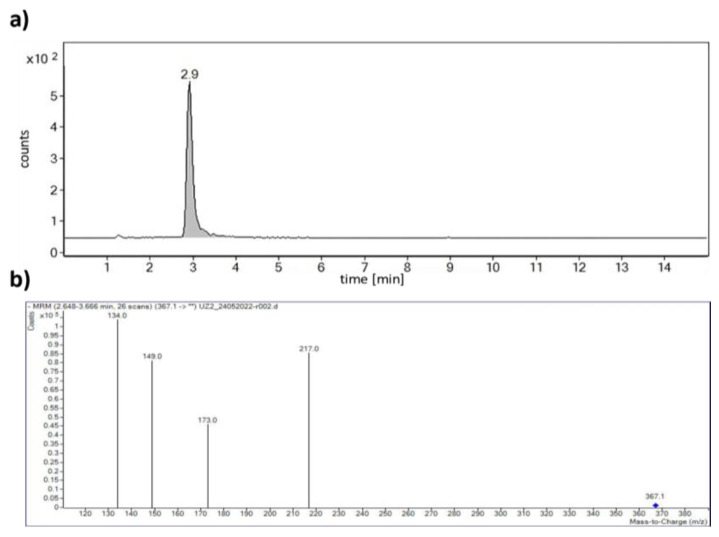
(**a**) Chromatogram of curcumin showing MRM spectrum in transition *m*/*z* 367.1 → 217.0, and (**b**) mass spectra of precursor [M − H]^−^ and product ions of curcumin. ** means all fragments of parent ion 367.1 at collision energy 4 and 12 V.

**Figure 4 foods-12-04000-f004:**
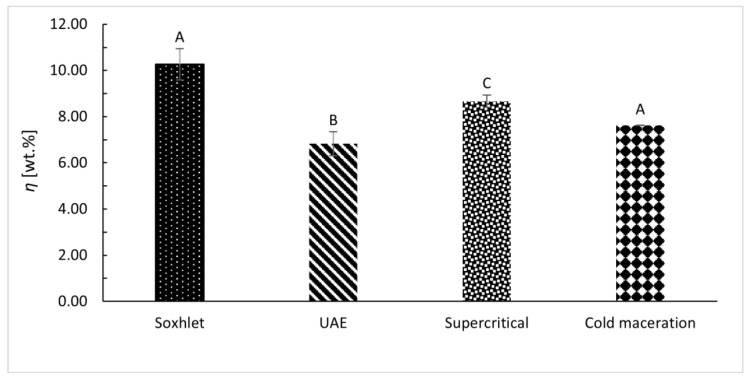
Comparison of different extraction methods (Soxhlet, UAE, supercritical, and cold maceration) on extraction yields (*η* wt.%). The error bars represent the standard deviation. Different letters indicate significant differences (*p* < 0.05) among different extraction methods.

**Figure 5 foods-12-04000-f005:**
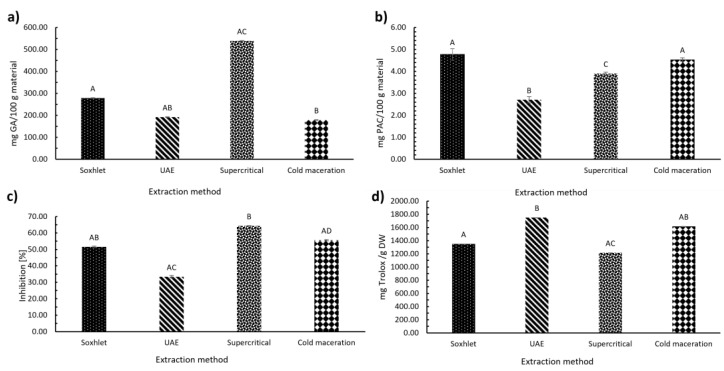
Characterisation of the extracts obtained by different extraction methods; (**a**) the total phenols content (in GA equivalents), (**b**) the proanthocyanidins content (PAC), (**c**) antioxidant activity determined based on DPPH^•^ method (in % inhibition), and (**d**) antioxidant activity determined based on ABTS^+•^ method (in Trolox equivalents). The error bars represent the standard deviations. Different letters indicate significant differences (*p* < 0.05) among different extraction methods.

**Figure 6 foods-12-04000-f006:**
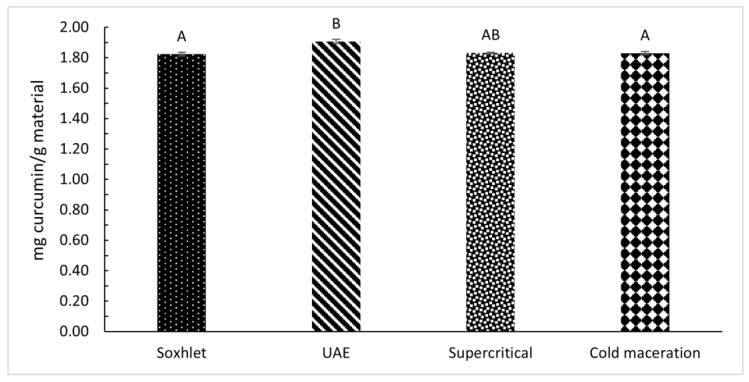
Content of curcumin in extracts using different extraction methods. The error bars represent the standard deviation. Different letters indicate significant differences (*p* < 0.05) among different extraction methods.

**Figure 7 foods-12-04000-f007:**
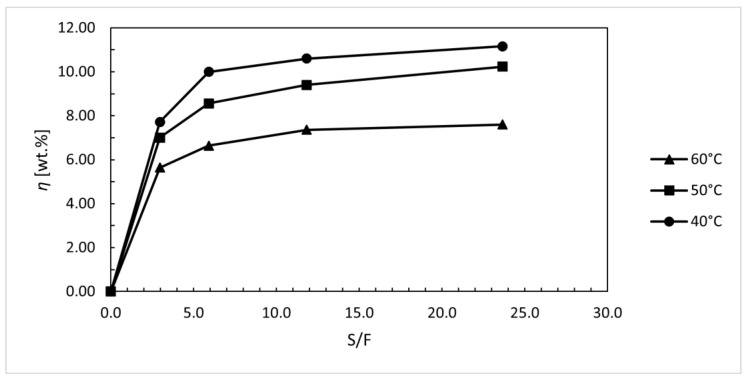
Extraction yield (wt.%) as a function of solvent-to-feed ratio (S/F) for supercritical extraction at pressure 25 MPa and temperatures 40, 50, and 60 °C.

**Figure 8 foods-12-04000-f008:**
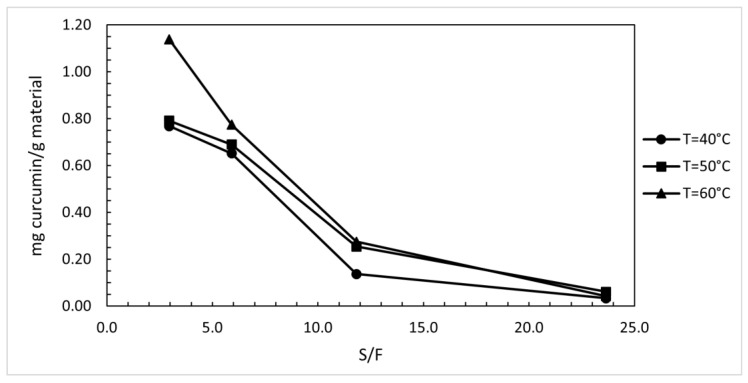
Curcumin content in the extract obtained by supercritical extraction at 25 MPa and 40, 50, 60 °C as a function of solvent-to-feed ratio.

## Data Availability

The original contributions presented in the study are included in the article; further inquiries can be directed to the corresponding author.

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
