# Peer review of "The Extraction Process, Separation, and Identification of Curcuminoids from Turmeric Curcuma longa"

_foods, 2023, doi:10.3390/foods12214000_

Round 1

Reviewer 1 Report

Comments and Suggestions for Authors

The comments for authors:

1. Overall conclusion is missing in the Abstract section.

2. The authors should check the text for typos.

3. What is the novelty and originality of this manuscript in comparison to previously published research? Include in the Introduction section.

4. Please, insert the extraction time in the section 2.3.1.4.

5. The authors should insert statistical analysis in Figure 3.

6. The term "optimization" should be removed from the title since the optimization has not been carried out.

Author Response

  1. Overall conclusion is missing in the Abstract section.

Answer: We appreciate the comment. We have added additional explanation in the Abstract.

  1. The authors should check the text for typos.

Answer:  We are thankful for such a close look. We have checked the text for typographical errors to the best of our knowledge.

  1. What is the novelty and originality of this manuscript in comparison to previously published research? Include in the Introduction section.

Answer: The article shows a comparison of different extraction techniques using organic solvents that have been proven the ability to recover high content of curcuminoids and supercritical fluid extraction. Supercritical fluid extraction has been applied as a sustainable method for the extraction of the target compounds. The novelty of the study is derived from the simultaneous investigation of the influence of extraction technique and process parameters (time, pressure, and temperature) on the content of bioactive components in turmeric extracts and subsequent statistical analysis. The results suggest that supercritical fluid extraction allowed easy separation of the solvent from the extracts and thus despite of lower mass yield, the recovery of phenolic compounds was highest. Additionally, the data on the curcumin extraction kinetics has been provided. Mass yield and curcumin content in the extract obtained by supercritical extraction at 25 MPa and 40, 50, and 60 °C as a function of solvent-to-feed ratio have been studied. Since the optimal process parameters and extraction time for the highest recovery of curcumin have been determined this data is perspective input for future studies.

  1. Please, insert the extraction time in the section 2.3.1.4.

Answer: We appreciate the comment. The extraction time has been provided in Section 2.3.1.4..

  1. The authors should insert statistical analysis in Figure 3.

Answer: We thank the reviewer for the comment. As suggested, statistical analysis was also performed for yield comparison. Data was distributed normally, and ANOVA was performed to compare the data. The obtained results were added to Figure 3.

  1. The term "optimization" should be removed from the title since the optimization has not been carried out.

Answer: We agree with the reviewer and the term ˝optimization˝ has been removed from the title and the new title with your suggestion is now ˝The extraction process, separation, and identification of curcuminoids from turmeric Curcuma longa˝.

Reviewer 2 Report

Comments and Suggestions for Authors

Dear authors the manuscript seems well structured and organised in its experimental parts although it lacks a biological component that could be supportive of the data obtained and which I would recommend considering. 

The following are my revisions and suggestions:

The sentence should be changed at lines 18-21 (The value of antioxidants activity...measureb by ABTS).

Please change ABTS and DPPH in abstract with ABTS+ and DPPHand in other parts of manuscript. 

Line 36-the name of familiy should be not in italic (Zingiberaceae).

Please, specify the standards that have been used for determination of total phenols and proanthocyanidins in Materials and Methods section.

I suggest to remove paragraph 2.4.3 and directly describe the ABTS and DPPH methods as paragraps 2.43 and 2.4.4, respectively.

I suggest to reduce the conclusions and summerise the results.  

Line 210- please change m/z with m/z

It is possible to reporte the data of DPPH and ABTS in the same way to have a comparison. 

Author Response

Dear authors the manuscript seems well structured and organised in its experimental parts although it lacks a biological component that could be supportive of the data obtained and which I would recommend considering.

The following are my revisions and suggestions:

The sentence should be changed at lines 18-21 (The value of antioxidants activity...measured by ABTS).

Answer: As suggested, the sentence in the abstract was changed ˝Antioxidant activity measured by DPPH was 64.27% and 1750.32 mg Trolox per g dry weight by ABTS+• for the extract obtained by supercritical extraction. ˝. (line 17-18)

Please change ABTS and DPPH in abstract with ABTS+• and DPPH• and in other parts of manuscript.

Answer: ABTS and DPPH have been corrected and changed throughout the manuscript with your corrections.

Line 36-the name of family should be not in italic (Zingiberaceae).

Answer: We appreciate such a close look. The correction was made accordingly.

Please, specify the standards that have been used for determination of total phenols and proanthocyanidins in Materials and Methods section.

Answer: We thank the reviewer for the comment.  All standards that have been used for the determination of total phenols and proanthocyanidins in the Materials and Methods section are provided in the revised manuscript.

I suggest to remove paragraph 2.4.3 and directly describe the ABTS and DPPH methods as paragraphs 2.4.3 and 2.4.4, respectively.

Answer: As you suggested, the paragraphs describing the ABTS and DPPH methods have been directly described as separate paragraphs 2.4.3. and 2.4.4. (line 190 and line 206).

I suggest to reduce the conclusions and summerise the results. 

Answer: The remark has been addressed, please see the marked version of the manuscript.

Line 210- please change m/z with m/z

Answer: We appreciate such a close look. The correction was made accordingly.

It is possible to report the data of DPPH and ABTS in the same way to have a comparison.

Answer: The antioxidant activity measurements can be performed using different mechanisms. Depending upon the reactions involved, these assays can roughly be classified into two types: assays based on hydrogen atom transfer (HAT) reactions and assays based on electron transfer (ET). The majority of HAT-based assays apply a competitive reaction scheme, in which antioxidant and substrate compete for thermally generated peroxyl radicals through the decomposition of azo compounds. ET based assays measure the capacity of an antioxidant in the reduction of an oxidant, which changes color when reduced. The degree of color change is correlated with the sample’s antioxidant concentrations. ET-based assays include the total phenols assay by Folin-Ciocalteu reagent and Trolox equivalence antioxidant capacity (TEAC or ABTS). The DPPH assay is technically simple, but some disadvantages limit its applications. Besides the mechanistic difference from the HAT reaction that normally occurs between antioxidants and peroxyl radicals, DPPH is a long-lived nitrogen radical, which bears no similarity to the highly reactive and transient peroxyl radicals involved in lipid peroxidation. According to different reactivity to Trolox using DPPH and ABTS, DPPH can not be expressed by Trolox equivalent. Just using one antioxidant activity method, the results can give wrong responses.

Reviewer 3 Report

Comments and Suggestions for Authors

The manuscript foods-2614196 deals with the results of a comparison of a series of differing extraction processes on Curcuma longa. The evaluation of yelds and antioxidant potency of the extract has been followeb by a standard colorimetric method (Folin), the radical scavenging properties have been evalkuated by DPPH and ABTS method, and curcumine determined by HPLC/MSMS method. Results have been also evaluated by ANOVA and  Kruskal-Wallis test.

The main point I raise however is that the samples that were indempndently extracted with each extracion method were just one per extraction, as it appears from the experimental description of the extraction processes. The variability of the results therefore is the result of the repeated measures made on each sample.

Given the aim of the authors this approach is too simplistic, as it removes the most important aspect to evaluate: variability (in yelds and other effects..) within and extraction processe (that could be easily expressed as standard deviation of the following measures). A single extraction experiment could be lucky or unfortunate enough to lead to misleading results. Multiple extractions (with the same protocol) of the raw material should be evaluated and averaged prior to express any evaluationon on the yelds and other aspects. A this point with a larger data set (with, likely, increased variaces among the variables) also other statistical analyses could be performed to (possibly) evidence the differences among extraction protocols. At the moment the assesment of signifcance of difference among differently extracted samples are likely due measurements variation... that are expltedly lower than that associated with multiple extractions.

If, instead, multiple extractions for each extraction technique have already been purposely made, it is non welll evidenced in the experimental parts, and it should.

Author Response

The manuscript foods-2614196 deals with the results of a comparison of a series of differing extraction processes on Curcuma longa. The evaluation of yelds and antioxidant potency of the extract has been followeb by a standard colorimetric method (Folin), the radical scavenging properties have been evalkuated by DPPH and ABTS method, and curcumine determined by HPLC/MSMS method. Results have been also evaluated by ANOVA and  Kruskal-Wallis test.

The main point I raise however is that the samples that were indempndently extracted with each extracion method were just one per extraction, as it appears from the experimental description of the extraction processes. The variability of the results therefore is the result of the repeated measures made on each sample.

Answer: We appreciate the comment. Every extraction procedure has been performed in triplicate and subsequent statistical analyses were performed as required. We have corrected the text to clearly report that such a procedure was employed. The standard deviations have been provided in the revised manuscript.

Given the aim of the authors this approach is too simplistic, as it removes the most important aspect to evaluate: variability (in yelds and other effects..) within and extraction processe (that could be easily expressed as standard deviation of the following measures).

Answer: As mentioned above, each extraction procedure has been performed in triplicate and standard deviations have been provided.

A single extraction experiment could be lucky or unfortunate enough to lead to misleading results. Multiple extractions (with the same protocol) of the raw material should be evaluated and averaged prior to express any evaluationon on the yelds and other aspects. A this point with a larger data set (with, likely, increased variaces among the variables) also other statistical analyses could be performed to (possibly) evidence the differences among extraction protocols. At the moment the assesment of signifcance of difference among differently extracted samples are likely due measurements variation... that are expltedly lower than that associated with multiple extractions.

If, instead, multiple extractions for each extraction technique have already been purposely made, it is non well evidenced in the experimental parts, and it should.

Answer: We appreciate the comment. We have performed additional statistical analyses and provided explanations based on the results obtained.

Reviewer 4 Report

Comments and Suggestions for Authors

The presented manuscript entitled “Optimization of the extraction process, separation, and identification of curcuminoids from turmeric Curcuma longa" concern the possibility of extracting curcuminoids form turmeric Curcuma longa with 3 conventional extraction techniques and supercitical fluid extraction. However, the research is not innovative and deep enough. Native speaker and several colleagues who are skilled authors to write the article must read your paper. 

1.The introduction should be rewrite. The aim of this manuscript is to study the influence of extraction technique and process parameters on the content of bioactive component in turmeic extracts. However, no content about different extraction method for curcuminoids extraction was mentioned in the introcuction.

2.Figure 2 is not clear enough, please revised it.

3.A representative chromatogram of HPLC-MS/MS analysis need to be provided.

4.Section 3.1, three parallel experiments are required.

5.Why used Kruska-Wallis statistical test instead of ANOVA in Figure 4a. Also different letters which represent the significant difference should be add to the figures.

6.For DPPH determination IC50 should be calculated.

7.To explain why DPPH and ABTS have opposite experimental results, it is recommended to carefully analyze the components in the extract using HPLC-MS/MS and explain the relevant results through solubility of different phenolic compounds.

8.Avoid using Fig.5 but Figure 5 in the manuscript. 

Comments on the Quality of English Language

 Native speaker and several colleagues who are skilled authors to write the article must read your paper. 

Author Response

The presented manuscript entitled “Optimization of the extraction process, separation, and identification of curcuminoids from turmeric Curcuma longa" concern the possibility of extracting curcuminoids form turmeric Curcuma longa with 3 conventional extraction techniques and supercitical fluid extraction. However, the research is not innovative and deep enough. Native speaker and several colleagues who are skilled authors to write the article must read your paper.

1.The introduction should be rewrite. The aim of this manuscript is to study the influence of extraction technique and process parameters on the content of bioactive component in turmeric extracts. However, no content about different extraction method for curcuminoids extraction was mentioned in the introduction.

Answer: We thank the reviewer for the comment. The content about extraction methods for extracting curcuminoids has been corrected and added to the introduction with your suggestion.

2.Figure 2 is not clear enough, please revised it.

Answer: We appreciate such a close look. The correction was made accordingly.

3.A representative chromatogram of HPLC-MS/MS analysis need to be provided.

Answer: The chromatogram of standard curcumin and sample analysed by HPLC-MS/MS is given in supplementary (Figure S1).

  1. a)

  1. b)

Figure S1: HPLC-MS/MS chromatogram of curcumin in a) standard solution and in b) cold maceration sample using ethanol for extraction.

4.Section 3.1, three parallel experiments are required.

Answer: We appreciate the comment. Every moisture measurement has been performed in triplicate We have corrected the text to clearly report that such a procedure was employed.

5.Why used Kruska-Wallis statistical test instead of ANOVA in Figure 4a. Also different letters which represent the significant difference should be add to the figures.

Answer: Kruskal-Wallis and ANOVA (Analysis of Variance) are both statistical tests used to analyse data involving multiple groups or treatments. However, they are applied in different situations based on the nature of our data and the assumptions that can be met. Here's when we would typically use each test:

  1. Kruskal-Wallis Test

   - Data Type: Kruskal-Wallis is a non-parametric test used when the data is ordinal, interval, or ratio, or it doesn't meet the assumptions of normality and homogeneity of variances required for ANOVA.

   - Groups: It's used when we have more than two independent groups (usually three or more) and we want to determine if there are significant differences among these groups.

   - Assumptions: It does not assume that the data is normally distributed or that the variances are equal across groups.

  1. ANOVA (Analysis of Variance):

   - Data Type: ANOVA is used when the data is continuous (interval or ratio) and the assumptions of normality and homogeneity of variances are met.

   - Groups: Like Kruskal-Wallis, ANOVA is used when there are three or more groups, and it tests if there are significant differences among the means of these groups.

   - Assumptions: ANOVA assumes that the data is normally distributed within each group and that the variances across groups are approximately equal.

ANOVA statistical test is only appropriate when the data is continuous, and assumptions are met. If ANOVA assumptions are violated, we are using non-parametric alternatives like Kruskal-Wallis. For each measured parameter statistical test for data distribution and homogeneity of variances was performed. In figure 4 the data did not follow a normal distribution and therefore Kruskal-Wallis test was selected.

Different letters which represent the significant differences between groups were added to the figures and the methods section was updated accordingly.

6.For DPPH determination IC50 should be calculated.

Answer: Thank you for suggestion, but in this study two antioxidant activity methods were used according to the applicability of these two methods on fat-soluble polyphenol compounds. As can be seen, the choice of proper AO method is in this case crucial. We are aware that DPPH results should be presented as IC50 concentrations, but in our case for better comparison the % of inhibition was more convenient. Please see the text in section 4 (Discussion) for additional explanation of the results.

7.To explain why DPPH and ABTS have opposite experimental results, it is recommended to carefully analyze the components in the extract using HPLC-MS/MS and explain the relevant results through solubility of different phenolic compounds.

Answer: Explanation is given in results.

8.Avoid using Fig.5 but Figure 5 in the manuscript.

Answer: We appreciate such a close look. The correction was made accordingly.

Comments on the Quality of English Language

Native speaker and several colleagues who are skilled authors to write the article must read your paper.

Answer: The manuscript has been now proof - checked by a native speaker.

Round 2

Reviewer 3 Report

Comments and Suggestions for Authors

From the emended article I see that the extraction procedures were repeated in triplicate. This gives more statistical value to the extraction procedures comparison. The article now stands on its own, even if, given the large amount of data on the extraction procedures on Curcuma longa, some perplexities on the novelty of the study still persist.

Author Response

After an extensive literature review, including more than 500 scientific original research manuscripts in Web of Sci, the authors concluded that the isolation of bioactive compounds from Curcuma longa is a topic of intense research. However, there is a lack of data on the comparison of different extraction techniques: in particular, most of the techniques used so far require the use of organic solvents. Supercritical extraction, which is described in some manuscripts, is also used, but there is virtually no data comparing the isolation of curcuminoids by different conventional organic solvent extraction methods. In addition, the analytical analysis of curcuminoids is a rather challenging task. Commonly used analytical technique is HIGH PERFORMANCE liquid chromatography combined with MS (9 results). However, this manuscript describes the use of high-performance liquid chromatography coupled to tandem mass spectrometry (HPLC-MS/MS). This method helps to separate the analyte(s) of interest from the components of the matrix, improves sensitivity and imprecision, and increases specificity by separating interferences. Only two results are available in the literature on this topic of curcuminoid analysis by HPLC-MS/MS. Finally, the authors state that there is no evidence of studies comparing different extraction techniques and the analytical performance of extracts by high-performance liquid chromatography coupled to tandem mass spectrometry (HPLC-MS/MS) in the literature. There is also no information on the evaluation of experimental extraction results by statistical analyses.

The following paragraph has been added to the introduction:
The aim of the manuscript is to compare different extraction techniques and the analytical performance of extracts by high-performance liquid chromatography coupled with tandem mass spectrometry (HPLC-MS/MS). In addition, the evaluation of experimental extraction results by statistical analyses were performed.